# Interpreting Adaptive Gradient Methods by Parameter Scaling for Learning-Rate-Free Optimization

## Abstract

We address the challenge of estimating the learning rate for adaptive gradient methods used in training deep neural networks. While several learning-rate-free approaches have been proposed, they are typically tailored for steepest descent. However, although steepest descent methods offer an intuitive approach to finding minima, many deep learning applications require adaptive gradient methods to achieve faster convergence. In this paper, we interpret adaptive gradient methods as steepest descent applied on parameter-scaled networks, proposing learning-rate-free adaptive gradient methods. Experimental results verify the effectiveness of this approach, demonstrating comparable performance to hand-tuned learning rates across various scenarios. This work extends the applicability of learning-rate-free methods, enhancing training with adaptive gradient methods.

## 1 Introduction

Almost all training procedures for deep neural networks are conducted iteratively, gradually approaching their optimum. During this process, the length and direction of parameter updates vary depending on the choice of optimizers, such as stochastic gradient descent (SGD), RMSProp (Tieleman & Hinton, 2012), and Adam (Kingma & Ba, 2015). In particular, adaptive gradient methods, such as RMSProp and Adam, are commonly employed in training sophisticated networks, including Transformers (Vaswani et al., 2017), which have recently gained significant prominence. These adaptive gradient methods dynamically adjust the direction of parameter updates by normalizing the gradient scale of each parameter. However, they still require the manual tuning of the learning rate, which determines the length of parameter updates, as a hyperparameter.

Since the learning rate hyperparameter significantly impacts the training speed and final performance of deep neural networks, extensive research has been conducted to adaptively determine learning rates (Orabona & Tommasi, 2017; Loizou et al., 2021; Ivgi et al., 2023; Defazio & Mishchenko, 2023; Mishchenko & Defazio, 2023), referred to as learning-rate-free methods. These methods estimate the learning rate based on the function value at the optimum, distance to the solution, or gradients. However, in the case of adaptive gradient methods, scaled gradients are used for training instead of the actual gradient values, making the estimation of the learning rate challenging.

To address this issue, methods for predicting the learning rate of adaptive gradient methods have also been proposed (Defazio & Mishchenko, 2023; Mishchenko & Defazio, 2023). However, it is observed that these methods do not consistently achieve optimal performance in certain situations, including the training of reinforcement learning models and fine-tuning on downstream tasks (Sec. 5). In this paper, we aim to address this issue and propose a learning rate estimation algorithm that can be effectively applied to adaptive gradient methods. We focus on Adam in our experiments as it is one of the most widely used methods, but the extension to others is straightforward.

Adaptive gradient methods scale the gradients of parameters to normalize their effects, resulting in a parameter update direction different from the steepest direction. To overcome this problem, we interpret an adaptive gradient method as a parameter-scaled SGD. As demonstrated in Eqs. 1-4, dividing the gradient by $\alpha^2$ is equivalent to multiplying the parameter by $\alpha$. Thus, adaptive gradient methods can be viewed as applying steepest descent to such parameter-scaled networks.

Table 1: Descriptions of adaptive gradients methods

| Method | Update rule | Reference |
|--------|-------------|-----------|
| AdaGrad | $w_{k+1} \leftarrow w_k - \eta g_k / \sqrt{\sum_{i \leq k} g_i^2}$ | Duchi et al. (2011) |
| RMSProp | $w_{k+1} \leftarrow w_k - \eta g_k / \sqrt{v_k}$ | Tieleman & Hinton (2012) |
| Adam | $w_{k+1} \leftarrow w_k - \eta m_k / \sqrt{v_k}$ | Kingma & Ba (2015) |
| AMSGrad | $w_{k+1} \leftarrow w_k - \eta m_k / \sqrt{\max_{i \leq k} v_i}$ | Reddi et al. (2018) |
| LARS | $w_{k+1} \leftarrow w_k - \eta m_k \phi(\|w_k\|) / \|m_k\|$ | You et al. (2017) |
| LAMB | $w_{k+1} \leftarrow w_k - \eta r_k \phi(\|w_k\|) / \|r_k\|; r_k = m_k / \sqrt{v_k}$ | You et al. (2020) |

$$y = f(w) = f(w'/\alpha) = f'(w') \tag{1}$$

$$w' \leftarrow w' - \eta \frac{\partial f'}{\partial w'}(w') \tag{2}$$

$$\alpha w \leftarrow \alpha w - \frac{\eta}{\alpha} \frac{\partial f}{\partial w}(w) \tag{3}$$

$$w \leftarrow w - \frac{\eta}{\alpha^2} \frac{\partial f}{\partial w}(w) \tag{4}$$

Based on this transformation, we propose two adaptive gradient variants of existing learning-rate-free methods (Loizou et al., 2021; Defazio & Mishchenko, 2023): parameter-scaled stochastic Polyak step-size (PS-SPS) and parameter-scaled D-Adapt SGD (PS-DA-SGD). The performance of existing and proposed learning-rate-free methods is validated under various experimental settings, including supervised classification, reinforcement learning, fine-tuning on downstream tasks, and self-supervised learning. While steepest descent-based learning-rate-free methods fail to converge in some cases, the proposed variants effectively transform them into adaptive gradient methods. Furthermore, one of the proposed methods, PS-DA-SGD, demonstrates the most robust and effective performance among the compared learning-rate-free methods.

The contributions of this paper can be summarized as follows:

1. We propose a method to interpret adaptive gradient methods as parameter-scaled SGD.

2. Using this transformation, we adapt existing learning-rate-free methods to be applicable to adaptive gradient methods.

3. We conduct a comprehensive evaluation of existing and proposed learning-rate-free adaptive gradient methods to assess their compatibility with various experimental scenarios.

## 2 RELATED WORK

### 2.1 ADAPTIVE GRADIENT METHODS

While steepest descent-based methods, such as SGD, SGD with momentum, and Nesterov accelerated gradient (Nesterov, 1983), offer an intuitive iterative approach to finding minima, experimental results indicate that adaptively adjusting the update direction leads to faster training of deep neural networks.

Table 1 provides descriptions of various adaptive gradient methods. Here, $m$ and $v$ represent the exponential moving average of gradients and squared gradients, respectively. For simplicity, we omit $\epsilon$ and bias correction terms. Most of these methods normalize the scale of each element in the gradient, though the specific approaches may differ. In this paper, our primary focus lies on the scaling rule of Adam, but it is worth noting that the extension to other methods is straightforward.

### 2.2 LEARNING-RATE-FREE METHODS

Convex optimization has long been a fundamental research area, and a key aspect of this field involves determining a learning rate that achieves optimal convergence rates. Theoretically, optimal

learning rate are known when problem-specific constants are available, such as the Lipschitz constant, distance to the solution, or function value at the optimum (Polyak, 1987; Duchi et al., 2011; Nesterov et al., 2018). However, as these values are problem-specific and often unknown, methods have been developed to adaptively find the learning rate without prior knowledge of these constants.

Although deep neural networks are typically not convex, the methods developed for convex optimization have proven to be effective in practice. In this paper, we focus on learning-rate-free methods that have demonstrated their effectiveness in deep learning applications. The methods that have been successfully applied in deep learning applications include continuous coin betting (CO-COB) (Orabona & Tommasi, 2017), stochastic Polyak step-size (SPS) (Loizou et al., 2021), distance over gradients (DoG) (Ivgi et al., 2023), and D-adaptation (Defazio & Mishchenko, 2023).

COCOB (Orabona & Tommasi, 2017) is a coin-betting-based method that considers situations where previous gradients remain relatively constant as indicative of a high expected outcome. In such cases, COCOB increases the betting, corresponding to an increase in the learning rate. To adapt this method for deep learning applications, a variant called COCOB-Backprop has also been proposed. This variant scales the learning rate for each parameter according to their gradient scales and has been validated for its effectiveness in image classification and word-level prediction tasks.

SPS (Loizou et al., 2021) is a stochastic variant of the Polyak step size (Polyak, 1987). It estimates the difference between the current function value and the optimum value, using the ratio of this difference to the square of the current gradient as the learning rate.

DoG (Ivgi et al., 2023) employs the displacement from the initial parameter as a proxy for the distance to the solution. The learning rate is determined by the ratio of the maximum displacement to the sum of the gradient norms up to the current iteration. The presence of the sum of gradient norms in the denominator of the learning rate formula gives DoG a learning rate annealing behavior similar to AdaGrad (Duchi et al., 2011).

D-Adaptation (Defazio & Mishchenko, 2023) estimates the distance to the solution by adaptively updating its lower bound during training. Several variants of D-Adaptation have been proposed, with D-Adapt SGD utilizing the lower bound divided by the gradient norm at the initial parameters as the learning rate. D-Adaptation-based methods can be combined with additional learning rate scheduling techniques, such as cosine annealing, to enhance convergence speed.

These learning-rate-free methods have demonstrated their effectiveness when applied to deep neural networks, often achieving equal or improved final accuracy compared to conventional optimizers. However, they are developed based on the assumption that gradient and parameter update directions align. Therefore, further validation is needed to assess their compatibility with adaptive gradient methods, which independently scale the gradients of each parameter.

## 3 MOTIVATION

The iterative process for finding the minimum of an objective function $f(w)$ can be written as follows:
$$w \leftarrow w - \eta u \tag{5}$$
Here, $\eta$ represents the learning rate, and $u$ denotes the update direction. Typically, $\eta$ is regarded as a hyperparameter that learning-rate-free methods aim to estimate, while $u$ is determined by the choice of optimizer.

In the case of SGD, the expectation value of $u$ is equal to the gradient of the objective function:
$$\text{SGD update rule:} \qquad \mathbb{E}u = \nabla f(w) \tag{6}$$
This holds true for momentum methods as well, provided that $f$ is smooth and the update are sufficiently small ($\|\eta u\| \ll 1/L$, when $\nabla f$ is $L$-Lipschitz continuous). For a more concise notation, we use $\nabla f(w)$ to represent $\frac{\partial f}{\partial w}(w)$.

However, this equality does not hold for adaptive gradient methods, where the gradient of each parameter is scaled:
$$\text{adaptive gradient:} \qquad \mathbb{E}u = \nabla f(w)/\alpha^2 \tag{7}$$
Here, $\alpha$ depends on the choice of scaling rule, with $\alpha^2 = \sqrt{v_t}$ being the case when Adam is used. Due to this discrepancy, the optimal $\eta$ of an adaptive gradient method should be a function of $\alpha$.

---

**Algorithm 1** Parameter-scaled stochastic Polyak step-size (PS-SPS)

---

Input: $f(\boldsymbol{w})$, $f^*$, $n$, $c$, $\boldsymbol{w}_0$, $\boldsymbol{\alpha}_k > 0$

1: **for** $k = 0$ **to** $n - 1$ **do**
2:     $\boldsymbol{g}_k \in \partial f(\boldsymbol{w}_k)$                                         ▷ Get stochastic gradients
3:     $\boldsymbol{w}_k' \leftarrow \boldsymbol{w}_k \circ \boldsymbol{\alpha}_k$                                    ▷ Scale parameters
4:     $\boldsymbol{g}_k' \leftarrow \boldsymbol{g}_k \circ \boldsymbol{\alpha}_k^{-1}$                                  ▷ Scale gradients
5:     $\eta_k \leftarrow \dfrac{f(\boldsymbol{w}_k) - f^*}{c\|\boldsymbol{g}_k'\|^2}$                                  ▷ SPS for SGD
6:     $\boldsymbol{w}_{k+1}' \leftarrow \boldsymbol{w}_k' - \eta_k \boldsymbol{g}_k'$                              ▷ Update parameters
7:     $\boldsymbol{w}_{k+1} \leftarrow \boldsymbol{w}_{k+1}' \circ \boldsymbol{\alpha}_k^{-1}$                          ▷ Undo parameter scaling
8: **end for**

---

Nevertheless, simply substituting the $\nabla f(w)$ term in learning-rate-free methods with $\nabla f(w)/\alpha^2$ results in an incorrect outcome.

For example, SPS estimates the learning rate as follows:

$$\eta = \frac{f(w) - f^*}{c\|\nabla f(w)\|^2} \tag{8}$$

However, replacing $\nabla f(w)$ with $\nabla f(w)/\alpha^2$ yields:

$$\eta(\alpha) = \frac{f(w) - f^*}{c\|\nabla f(w)/\alpha^2\|^2} \tag{9}$$

$$\mathbb{E}u(\alpha) = \nabla f(w)/\alpha^2 \tag{10}$$

$$\mathbb{E}\eta(\alpha)u(\alpha) = \frac{f(w) - f^*}{c\|\nabla f(w)\|^2}\nabla f(w) \cdot \alpha^2 \tag{11}$$

If $\eta(\alpha)$ was an optimal learning rate of the $\alpha^2$-scaled adaptive gradient method, the parameter update should remain consistent regardless of $\alpha$, but this is not the case. Hence, an alternative approach is needed to make learning-rate-free methods compatible with adaptive gradient methods.

In contrast, DoG and D-Adaptation produce parameter update rules that do not rely on $\alpha$ when $\alpha$ is a scalar. However, we observe they exhibit suboptimal performance when $\alpha$ takes the form of a vector with varying values for each component, which is the case for most adaptive gradient methods.

**Why are adaptive gradient methods important?** While SGD with momentum performs well in many scenarios, adaptive gradient methods tend to outperform plain SGD in scenarios involving sophisticated networks like Transformers or in scenarios where batch normalization cannot be applied, such as reinforcement learning (Bhatt et al., 2019; Zhang et al., 2020).

**Why is the proposed parameter scaling method important?** Adaptive gradient methods, despite scaling the gradient for each parameter, still necessitate the use of a learning rate hyperparameter. Moreover, the choice of the learning rate value has a significant impact on convergence speed and final performance. The proposed parameter scaling method serves to adapt learning-rate-free methods originally designed for SGD to be compatible with adaptive gradient methods.

## 4 PROPOSED METHOD

### 4.1 ALGORITHM

In this section, we introduce adaptive gradient variants by leveraging the parameter scaling technique. Specifically, we present parameter-scaled (PS) variants of SPS and D-Adaptation, denoted as PS-SPS and PS-DA, respectively. We omit COCOB (Orabona & Tommasi, 2017) and DoG (Ivgi et al., 2023) from consideration, as we find that PS-SPS and PS-DA perform well in practice, while the PS variant of DoG does not yield favorable results. Additionally, since COCOB is inherently a parameter-wise method, applying parameter scaling to it would be irrelevant.

---

**Algorithm 2** Parameter-scaled D-Adapt SGD (PS-DA-SGD)

Input: $f(\boldsymbol{w})$, $n$, $d_0$, $\gamma_k$, $\boldsymbol{w}_0$, $\boldsymbol{\alpha}_k > 0$
1: $m_0 = 0$, $\boldsymbol{s}_0 = \boldsymbol{0}$, $\boldsymbol{g}_M = \boldsymbol{0}$, $\boldsymbol{\alpha}_M = \boldsymbol{0}$
2: **for** $k = 0$ **to** $n - 1$ **do**
3: $\quad \boldsymbol{g}_k \in \partial f(\boldsymbol{w}_k)$ $\qquad\qquad\qquad\qquad\qquad\qquad$ ▷ Get stochastic gradients
4: $\quad \boldsymbol{w}'_k \leftarrow \boldsymbol{w}_k \circ \boldsymbol{\alpha}_k$; $\boldsymbol{w}'_0 \leftarrow \boldsymbol{w}_0 \circ \boldsymbol{\alpha}_k$ $\qquad\qquad\qquad$ ▷ Scale parameters
5: $\quad \boldsymbol{g}'_k \leftarrow \boldsymbol{g}_k \circ \boldsymbol{\alpha}_k^{-1}$; $\boldsymbol{s}'_k \leftarrow \boldsymbol{s}_k \circ \boldsymbol{\alpha}_k^{-1}$ $\qquad\qquad\quad$ ▷ Scale gradients
6: $\quad \boldsymbol{\alpha}_M \leftarrow \max(\boldsymbol{\alpha}_M, \boldsymbol{\alpha}_k)$ $\qquad\qquad\quad$ ▷ Update maximum scaling factor
7: $\quad d'_k \leftarrow d_k \cdot \max(\boldsymbol{\alpha}_k / \boldsymbol{\alpha}_M)$ $\qquad\qquad\qquad$ ▷ Scale D lower bound
8: $\quad \boldsymbol{g}_M \leftarrow \max(\boldsymbol{g}_M, \boldsymbol{g}_k)$ $\qquad\qquad\qquad$ ▷ Update maximum gradients
9: $\quad \boldsymbol{g}'_M \leftarrow \boldsymbol{g}_M \circ \boldsymbol{\alpha}_M^{-1}$ $\qquad\qquad\qquad\quad$ ▷ Scale maximum gradients
10: $\quad \eta_k \leftarrow \dfrac{d'_k \gamma_k}{\|\boldsymbol{g}'_M\|}$ $\qquad\qquad\qquad\qquad$ ▷ D-Adapt SGD learning rate
11: $\quad \boldsymbol{w}'_{k+1} \leftarrow \boldsymbol{w}'_k - \eta_k \boldsymbol{g}'_k$ $\qquad\qquad\qquad\quad$ ▷ Update parameters
12: $\quad \hat{d}_{k+1} \leftarrow \dfrac{m_k}{\|\boldsymbol{s}'_k\|}$ $\qquad\qquad\qquad\qquad$ ▷ Get D lower bound
13: $\quad d_{k+1} \leftarrow \max(\hat{d}_{k+1}, d_k)$ $\qquad\qquad\qquad$ ▷ Update D lower bound
14: $\quad m_{k+1} \leftarrow m_k + \eta_k \langle \boldsymbol{g}'_k, \boldsymbol{w}'_0 - \boldsymbol{w}'_k \rangle$ $\qquad\qquad$ ▷ Update numerator
15: $\quad \boldsymbol{s}_{k+1} \leftarrow \boldsymbol{s}_k + \eta_k \boldsymbol{g}_k$ $\qquad\qquad\qquad$ ▷ Update denominator
16: $\quad \boldsymbol{w}_{k+1} \leftarrow \boldsymbol{w}'_{k+1} \circ \boldsymbol{\alpha}_k^{-1}$ $\qquad\qquad\quad$ ▷ Undo parameter scaling
17: **end for**

---

The conversion of SPS to PS-SPS is straightforward, as illustrated in Alg. 1.

Here, $\circ$ denotes element-wise multiplication, and $\boldsymbol{\alpha}_k^{-1}$ signifies the element-wise inverse of $\boldsymbol{\alpha}_k$. $f^*$ and $c$ represent the function value at optimum and the scaling parameter, respectively, which are essential for estimating the learning rate using SPS. We employ $c = 0.5$ throughout all experiments, following Loizou et al. (2021). $\boldsymbol{\alpha}_k$ represents the scaling factor, which varies depending on the choice of an adaptive gradient method. To convert SPS to PS-SPS, we only need to scale $\boldsymbol{g}_k$ and $\boldsymbol{w}_k$, as the only affected term in SPS is $\boldsymbol{g}_k$.

On the other hand, the conversion of D-Adapt SGD to PS-DA-SGD requires several modifications, as described in Alg. 2.

$\max(\boldsymbol{x}, \boldsymbol{y})$ represents the element-wise maximum of two vectors, and $\max(\boldsymbol{x})$ signifies the maximum element within a vector. $d_0$ stands for the initial estimate of the lower bound of $D$, which represents the distance to the solution. Consistent with Defazio & Mishchenko (2023), a value of $d_0 = 10^{-6}$ is employed in all experiments. Additionally, $\gamma_k$ denotes the learning rate annealing schedule that can be applied in conjunction with D-Adaptation.

We need additional modifications to convert D-Adapt SGD to PS-DA-SGD, because applying the modifications outlined in Alg. 1 directly to D-Adapt SGD produce undesired results. This is because the learning rate employed by D-Adapt SGD depends on the ratio of $D$ to $\boldsymbol{g}_0$, where $\boldsymbol{g}_0$ represents the gradient at initial parameters. If one of the parameters converges faster than the others, the element of $\boldsymbol{\alpha}_k$ corresponding to that parameter may approach zero, especially when an Adam-style scaling rule is applied. This results in an increase of in $\boldsymbol{g}_0 \circ \boldsymbol{\alpha}_k^{-1}$, and consequently, the training ceases. Therefore, we introduce additional modifications based on following two key insights: firstly, the update rule should remain consistent even when an alpha with all equal elements is multiplied to all parameters, and secondly, even if one parameter converges earlier, the remaining parameters should continue their training. To compensate for the decrease of $\boldsymbol{\alpha}$, we shrink $D$ by the maximum scale ratio (Lines 6-7). Additionally, we use $\boldsymbol{g}_M$ instead of $\boldsymbol{g}_0$ and scale $\boldsymbol{g}_M$ by $\boldsymbol{\alpha}_M$ instead of $\boldsymbol{\alpha}_k$ to prevent training from halting (Line 8-9).

## 4.2 CONVERGENCE ANALYSIS

Reddi et al. (2018) demonstrated that Adam may fail to converge, even when applied to a convex function. To address this issue, they introduced AMSGrad, which they proved to be convergent. Adam scales gradients based on to the square root of moving average of squared gradients, which

Table 2: Descriptions of learning-rate-free and baseline methods

| Method | Note | Reference |
|--------|------|-----------|
| *Steepest descent* | | |
| SGD | (Baseline) requires a hand-tuned learning rate | |
| SPS | Estimated learning rate fluctuates | Loizou et al. (2021) |
| DoG | Implicit AdaGrad-like learning rate annealing is inherent | Ivgi et al. (2023) |
| D-Adapt SGD | SGD variant of D-Adaptation
Additional learning rate annealing can be applied | Defazio & Mishchenko (2023) |
| *Adaptive gradient* | | |
| Adam | (Baseline) requires a hand-tuned learning rate | Kingma & Ba (2015) |
| COCOB | Estimates parameter-wise learning rates | Orabona & Tommasi (2017) |
| LDoG | Layer-wise variant of DoG | Ivgi et al. (2023) |
| D-Adapt Adam | Adam variant of D-Adaptation | Defazio & Mishchenko (2023) |
| Prodigy | Adam variant of D-Adaptation with faster adaptation | Mishchenko & Defazio (2023) |
| *Proposed* | | |
| PS-SPS | Proposed parameter-scaled SPS | |
| PS-DA-SGD | Proposed parameter-scaled D-Adapt SGD | |

can decrease as the training progresses. Conversely, AMSGrad uses the maximum scaling of Adam up to the current iteration, preventing the scaling from decreasing.

Similarly, the convergence of both proposed algorithm depends on the parameter scaling rule ($\alpha_k$). When the AMSGrad scaling rule is applied, both methods find the optimum of a convex Lipschitz function, as demonstrated in Appendix A.1.

However, in practice, the scaling rule of AMSGrad often exhibits slower convergence compared to Adam. Consequently, we employ Adam scaling throughout our experiments.

## 5 EXPERIMENTS

In this section, we present experimental results to demonstrate the necessity of our proposed method. We begin by examining the failure cases of existing methods and then validate the performance of our approach by comparing it to existing methods. Throughout the experiments, we apply Adam parameter scaling rule to both PS-SPS and PS-DA-SGD. Additionally, we apply a momentum of 0.9 to PS-DA-SGD.

### 5.1 OPTIMIZERS

Table 2 presents the optimization methods we employ for comparison. We classify COCOB as an adaptive gradient method since it estimates parameter-wise learning rates, and similarly, we classify LDoG as an adaptive gradient method because it estimates layer-wise learning rates.

### 5.2 TESTING ENVIRONMENTS

We begin by examining the impact of batch normalization layers (Ioffe & Szegedy, 2015) on the performance of learning-rate-free methods. To assess this, we train ResNet-32 (He et al., 2016) and VGG-19 (Simonyan & Zisserman, 2015) models on the CIFAR-100 dataset (Krizhevsky et al., 2009), removing all batch normalization layers from the VGG models while retaining them in the ResNet models.

Next, we assess performance in reinforcement learning tasks. Due to the non-stationary nature of the task, batch normalization layers exhibit inferior performance and are consequently omitted from this experiment. We train soft actor-critic (SAC; Haarnoja et al. (2018)) models on the Hopper-v2 and Humanoid-v2 benchmarks from the OpenAI gym benchmark suite (Brockman et al., 2016).

Table 3: Experimental results. Cases achieving at least $95\%$ of the best results are shown in bold.

| | CIFAR-100 | | Reinforcement learning | | ImageNet | Fine-tuning | | SSL |
|---|---|---|---|---|---|---|---|---|
| | ResNet-32 | VGG-19 | Hopper | Humanoid | ViT-Tiny | SegNeXt | Swin-T | SimCLR |
| *Steepest descent* | | | | | | | | |
| SGD | 66.7 (0.233) | 64.7 (0.038) | 3466 (5/5) | NaN (0/5) | - | - | - | 74.9 |
| SPS | **65.2 (0.359)** | 0.01 (5.041) | - | - | 18.9 | **39.7** | 41.4 | **76.0** |
| DoG | **66.7 (0.495)** | 56.3 (0.074) | 1050 (0/5) | 475 (0/5) | 0.31 | 39.0 | **43.0** | 75.7 |
| D-Adapt SGD | **65.6 (0.312)** | 0.01 (4.605) | NaN (0/5) | NaN (0/5) | 0.77 | 38.1 | **42.0** | **74.7** |
| *Adaptive gradient* | | | | | | | | |
| Adam | 68.3 (0.236) | 62.1 (0.086) | 3448 (5/5) | 5527 (5/5) | 72.0 | 41.6 | 44.1 | 76.0 |
| COCOB | 50.1 (1.666) | 55.1 (0.771) | **3334 (5/5)** | **5337 (5/5)** | 0.14 | 6.12 | 18.6 | 66.4 |
| LDoG | 63.9 (0.547) | 56.4 (0.073) | 3277 (1/5) | 469 (0/5) | 4.29 | **40.6** | **43.4** | 74.5 |
| D-Adapt Adam | **65.9 (0.286)** | 48.3 (0.033) | **3502 (5/5)** | 5504 (3/5) | **70.5** | 26.1 | 41.4 | 69.0 |
| Prodigy | **66.4 (0.221)** | 57.8 (0.019) | 195 (0/5) | 419 (0/5) | 68.3 | 29.3 | 41.8 | **76.9** |
| *Proposed* | | | | | | | | |
| PS-SPS | **65.9 (0.337)** | 53.4 (0.182) | - | - | 42.0 | 39.3 | **43.8** | **76.7** |
| PS-DA-SGD | **65.8 (0.305)** | 50.6 (0.009) | **3497 (5/5)** | **5534 (5/5)** | **73.6** | **40.6** | **44.0** | 73.5 |

Subsequently, we train Vision Transformer (ViT) models (Dosovitskiy et al., 2021) on the ImageNet-1K dataset (Russakovsky et al., 2015). This model demands the use of adaptive gradient methods to achieve a faster convergence.

We also conduct two additional experiments to assess the performance of learning-rate-free methods. One of these experiments involves fine-tuning models with pretrained weights. We fine-tune two semantic segmentation networks on the ADE20K dataset (Zhou et al., 2019): SegNeXt (Guo et al., 2022), a convolution-based model, and Swin Transformer (Liu et al., 2021), a Transformer-based model.

The second additional experiment involves a self-supervised learning (SSL) task. We train Sim-CLR models (Chen et al., 2020) with a ResNet-18 backbone (He et al., 2016) on the STL-10 dataset (Coates et al., 2011).

## 5.3 EXPERIMENTAL RESULTS

Table 3 summarizes the experiment results. We opt not to employ the weight decay regularization for the sake of a fair comparison. It is because the selection of its magnitude can significantly influence the final performance, with the optimal magnitude varying depending on optimization methods. The CIFAR-100 column displays the top-1 accuracy on the test set. We also provide the train loss in parentheses to distinguish overfitting. The reinforcement learning column displays average reward. Because reinforcement learning models sometimes fail to converge depending on their initial parameters or optimization algorithm, we also provide the training success rate in parentheses. The last three columns represent the top-1 accuracy on ImageNet, the mean intersection-over-union on ADE20K, and the top-1 accuracy on STL-10, respectively.

The proposed parameter scaling approach successfully converts existing steepest descent-based learning-rate-free methods to suitable for adaptive gradient methods, from SPS to PS-SPS and D-Adapt SGD to PS-DA-SGD, enabling them to achieve convergence on a wider range of tasks. Notably, PS-DA-SGD performs exceptionally well, surpassing existing learning-rate-free methods on most tasks and closing the gap with hand-tuned learning rates.

## 6 ANALYSIS

### 6.1 SUPERVISED CLASSIFICATION ON CIFAR-100

All the methods, except for COCOB, demonstrate reasonable performance on ResNet-32 models. However, the performance of steepest descent methods deteriorate when applied to VGG-19 models without batch normalization layers. Amongst the steepest descent-based learning-rate-free methods, only DoG converges, whereas all adaptive gradient methods succeed.

It is observed that learning-rate-free methods tend to exhibit lower accuracy in VGG experiments, primarily due to overfitting. This phenomenon can be attributed to the fact that learning-rate-free

Table 4: ImageNet top-1 accuracy of ViT with weight decay regularization

| Method | AdamW | D-Adapt Adam | Prodigy | PS-DA-SGD |
|---|---|---|---|---|
| **Top-1 acc.** | 75.4 | 72.4 | 74.6 | 75.0 |

methods have developed from convex optimization, where the primary goal is to minimize loss on the training dataset. In contrast, deep learning applications also require generalization capabilities.

## 6.2 REINFORCEMENT LEARNING

The performance of steepest descent-based methods tends to worsen in reinforcement learning models, where batch normalization is often omitted. All steepest descent-based learning-rate-free methods fail to converge. Moreover, several adaptive gradient-based methods also exhibit decreased or unstable performance. Nonetheless, the proposed PS-DA-SGD achieves stable and superior performance. SPS and PS-SPS are excluded since they require the loss value at the optimum, denoted as $f^*$ in Alg. 1, which is unavailable in the reinforcement learning experiments.

Similar to the proposed PS-DA-SGD, D-Adapt Adam and Prodigy are both variants of Adam based on D-Adaptation. However, they exhibit unstable performance in reinforcement learning. Estimated learning rates can help explain this phenomenon, as depicted in Fig. 1a. In all trials, the estimated learning rates of Prodigy diverge, and occasionally, the learning rates of D-Adapt Adam also exhibit divergence. The design goal of Prodigy is to adapt the learning rate quickly, suspected to be the cause of divergence.

## 6.3 TRAINING TRANSFORMER FROM SCRATCH

Again, all steepest gradient-based methods diverge or fail to reach the optimum. Most adaptive gradient-based methods are successful, but the final performance varies depending on each method. D-Adapt Adam and PS-DA-SGD show the best performance among learning-rate-free methods, comparable to that of Adam. PS-SPS underperforms in the ViT experiment compared to other experiments due to the highly fluctuating learning rate. This is also true for PS-SPS, which is a variant of SPS, and the fluctuation hinders the training speed of PS-SPS. However, we observe that the training of PS-SPS is slow but still ongoing, and therefore, we anticipate that it will eventually converge to a solution given sufficient time.

While we omit weight decay regularization throughout the experiments, other methods reported their performance on ViT with weight decay. For a fair comparison, we also train ViT using PS-DA-SGD with weight decay applied and compared it to those methods. As shown in Table 4, the performance gap between methods decrease when weight decay is enabled. Nevertheless, PS-DA-SGD demonstrates the best accuracy among learning-rate-free methods.

## 6.4 FINE-TUNING FROM PRETRAINED WEIGHTS

All methods, except for COCOB, exhibit reasonable performance in fine-tuning experiments. However, similar to the reinforcement learning experiments, D-Adapt Adam and Prodigy show suboptimal performance. Fig. 1b displays the estimated learning rates. D-Adapt Adam and Prodigy predict relatively large learning rates, which result in suboptimal performance on the fine-tuning task.

## 6.5 SELF-SUPERVISED LEARNING

We expected self-supervised learning task to be challenging to optimize because there is no a supervision or defined optimal goal. However, contrary to expectations, most methods perform well in self-supervised learning. Prodigy and PS-SPS outperform D-Adapt Adam and PS-DA-SGD on this task, primarily due to their larger learning rates. Fig. 1c shows that D-Adapt Adam and PS-DA-SGD estimate lower learning rates compared to Prodigy and PS-SPS. We suspect that this may also be due to overfitting, as STL-10 is a relatively small dataset, similar to CIFAR-100. Because a large learning rate can act as a regularizer (Li et al., 2019), Prodigy and PS-SPS achieve better test accuracy in the self-supervised learning experiment.

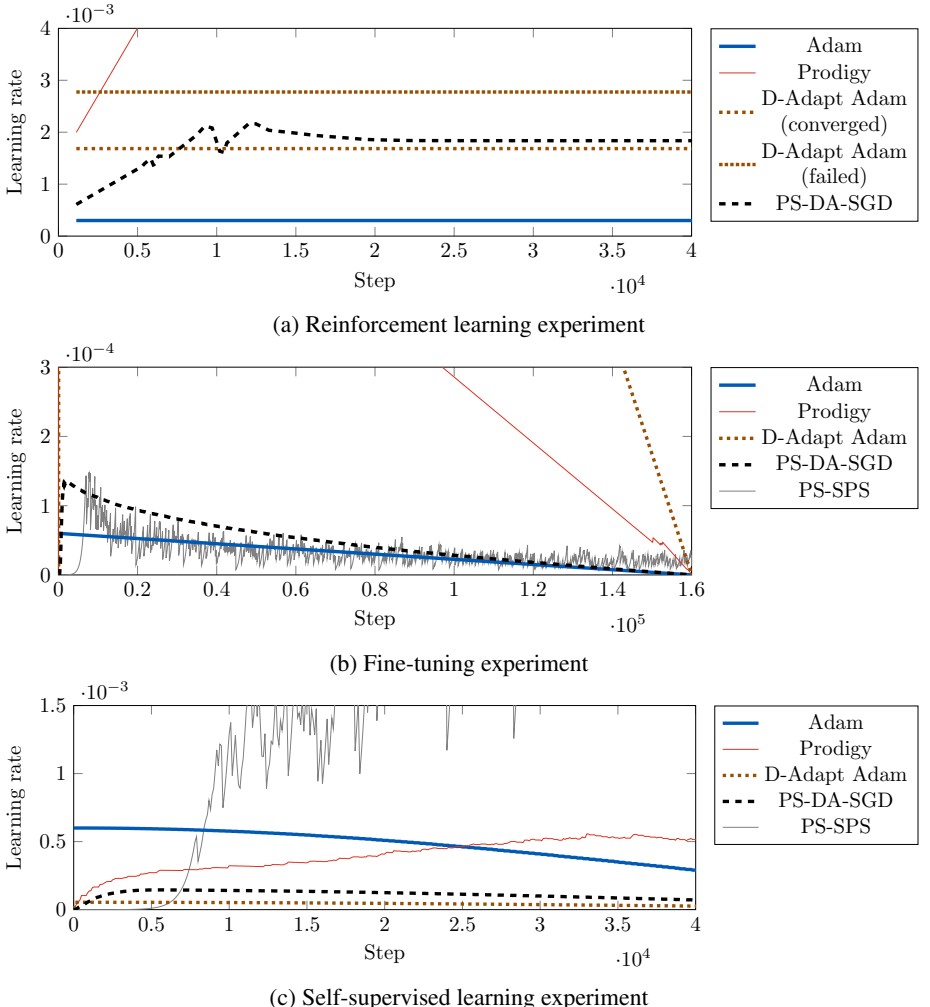

(a) Reinforcement learning experiment

(b) Fine-tuning experiment

(c) Self-supervised learning experiment

Figure 1: Estimated learning rate

## 6.6 LIMITATION

Learning-rate-free methods tend to suffer from overfitting more severely than hand-tuned ones, especially on small datasets. We anticipate that this is due to the disparity in goals between convex optimization and deep learning applications, as deep learning applications also require generalization capabilities. While adding regularization like weight decay can help mitigate this issue, it introduces another hyperparameter to tune, which contradicts the objective of learning-rate-free optimization. Therefore, further research is needed to incorporate regularization or sharpness-aware minimization (Foret et al., 2021) to address this issue.

## 7 CONCLUSION

In this paper, we demonstrate that steepest descent-based learning-rate-free methods encounter challenges in specific scenarios. To address this issue, we introduce two adaptive gradient variants of these methods, namely PS-SPS and PS-DA-SGD. These variants are founded on the insight that an adaptive gradient can be viewed as equivalent to a steepest descent method applied to a parameter-scaled network. Our proposed approach effectively transforms existing steepest descent-based methods into adaptive gradient methods, enabling them to achieve convergence across a wider range of tasks. Furthermore, our proposed methods exhibit the highest levels of stability and efficiency among adaptive gradient-based learning-rate-free methods.

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

# A APPENDIX

## A.1 CONVERGENCE ANALYSIS OF PROPOSED METHODS

As mentioned in Sec. 4.2, the convergence of adaptive gradient methods depends on the scaling rule. In this section, we analyze the convergence of proposed methods in the case of AMSGrad scaling. We also assume that a convex $G$-Lipschitz function $f$ and the deterministic case. As shown in Table 1, the scaling rule of AMSGrad is $\alpha_k^2 = \sqrt{\max_{i \leq k} v_i}$. Because $\alpha_k$ is a non-decreasing sequence upper bounded by $\sqrt{G}$, it converges to $\alpha$. As the scaling rule converges, the convergence of proposed methods are straightforwardly derived from their vanilla ones (Loizou et al., 2021; Defazio & Mishchenko, 2023).

### A.1.1 CONVERGENCE ANALYSIS OF PS-SPS

In the deterministic case, we use $c = 1$ in the Alg. 1. From the parameter update of PS-SPS, we derive the following:

$$\|(\boldsymbol{w}_{k+1} - \boldsymbol{w}^*) \circ \boldsymbol{\alpha}_k\| = \|(\boldsymbol{w}_k - \boldsymbol{w}^*) \circ \boldsymbol{\alpha}_k - \eta_k \boldsymbol{g}_k \circ \boldsymbol{\alpha}_k^{-1}\| \tag{12}$$

Using the convexity of $f$ and applying the definition of $\eta_k$ in the Alg. 1 yields:

$$\begin{aligned}
&\|(\boldsymbol{w}_{k+1} - \boldsymbol{w}^*) \circ \boldsymbol{\alpha}_k\|^2 \\
&= \|(\boldsymbol{w}_k - \boldsymbol{w}^*) \circ \boldsymbol{\alpha}_k\|^2 + \eta_k^2 \|\boldsymbol{g}_k \circ \boldsymbol{\alpha}_k^{-1}\|^2 - 2\eta_k \langle \boldsymbol{w}_k - \boldsymbol{w}^*, \boldsymbol{g}_k \rangle \\
&\leq \|(\boldsymbol{w}_k - \boldsymbol{w}^*) \circ \boldsymbol{\alpha}_k\|^2 + \eta_k^2 \|\boldsymbol{g}_k \circ \boldsymbol{\alpha}_k^{-1}\|^2 - 2\eta_k (f(\boldsymbol{w}_k) - f(\boldsymbol{w}^*)) \\
&= \|(\boldsymbol{w}_k - \boldsymbol{w}^*) \circ \boldsymbol{\alpha}_k\|^2 - \frac{(f(\boldsymbol{w}_k) - f(\boldsymbol{w}^*))^2}{\|\boldsymbol{g}_k \circ \boldsymbol{\alpha}_k^{-1}\|^2} \\
&\leq \max(\boldsymbol{\alpha}_k \circ \boldsymbol{\alpha}_{k-1}^{-1})^2 \|(\boldsymbol{w}_k - \boldsymbol{w}^*) \circ \boldsymbol{\alpha}_{k-1}\|^2 - \frac{(f(\boldsymbol{w}_k) - f(\boldsymbol{w}^*))^2}{\|\boldsymbol{g}_k \circ \boldsymbol{\alpha}_k^{-1}\|^2}
\end{aligned} \tag{13}$$

Because $\boldsymbol{\alpha}_k$ converges to $\boldsymbol{\alpha}$, $\boldsymbol{\alpha}_k > \boldsymbol{\alpha}/2$ for $k > m$.

$$\begin{aligned}
&\|(\boldsymbol{w}_{n+1} - \boldsymbol{w}^*) \circ \boldsymbol{\alpha}_n\|^2 \\
&\leq \left( \prod_{k=m+1}^{n} \max(\boldsymbol{\alpha}_k \circ \boldsymbol{\alpha}_{k-1}^{-1})^2 \right) \|(\boldsymbol{w}_{m+1} - w^*) \circ \boldsymbol{\alpha}_m\|^2 \\
&\quad - \sum_{k=m+1}^{n} \left( \prod_{i=k+1}^{n} \max(\boldsymbol{\alpha}_i \circ \boldsymbol{\alpha}_{i-1}^{-1})^2 \right) \frac{(f(\boldsymbol{w}_k) - f(\boldsymbol{w}^*))^2}{\|\boldsymbol{g}_k \circ \boldsymbol{\alpha}_k^{-1}\|^2}
\end{aligned} \tag{14}$$

Suppose $\beta_k$ is the product of all elements of $\boldsymbol{\alpha}_k$. Then,

$$\max(\boldsymbol{\alpha}_k \circ \boldsymbol{\alpha}_{k-1}^{-1}) \leq \beta_k \cdot \beta_{k-1}^{-1}$$

$$1 \leq \prod_{k=m+1}^{n} \max(\boldsymbol{\alpha}_k \circ \boldsymbol{\alpha}_{k-1}^{-1}) \leq \beta_n \cdot \beta_m^{-1} = A^2 < \infty \tag{15}$$

Therefore,

$$0 \leq \|(\boldsymbol{w}_{n+1} - \boldsymbol{w}^*) \circ \boldsymbol{\alpha}_n\|^2 \leq A^2 \|(\boldsymbol{w}_{m+1} - w^*) \circ \boldsymbol{\alpha}_m\|^2 - \sum_{k=m+1}^{n} \frac{(f(\boldsymbol{w}_k) - f(\boldsymbol{w}^*))^2}{\|\boldsymbol{g}_k \circ \boldsymbol{\alpha}_k^{-1}\|^2}$$

$$\leq A^2 D^2 G - \sum_{k=m+1}^{n} (f(\boldsymbol{w}_k) - f(\boldsymbol{w}^*))^2 G^{-2} \min(\boldsymbol{\alpha})^2/4 \tag{16}$$

$$\min_{m+1 \leq k \leq n} f(\boldsymbol{w}_k) - f(\boldsymbol{w}^*) \leq \frac{2ADG^{3/2} \min(\boldsymbol{\alpha})^{-1}}{\sqrt{n - m - 1}} \tag{17}$$

### A.1.2 Convergence analysis of PS-DA-SGD

D-Adaptation-based methods contain additional learning rate annealing parameter, $\gamma_k$, and the convergence of PS-DA-SGD depends on it. PS-DA-SGD converges when $\gamma_k$ is a decreasing sequence that satisfying

$$\sum_{k=1}^{\infty} \gamma_k = \infty, \sum_{k=1}^{\infty} \gamma_k^2 < \infty. \tag{18}$$

At first, we need that $d_k$ converges. From the convexity of $f$,

$$
\begin{aligned}
0 &\le \sum_{k=0}^{n} \eta_k (f(\boldsymbol{w}_k) - f(\boldsymbol{w}^*)) \\
&\le \sum_{k=0}^{n} \langle \boldsymbol{g}_k, \boldsymbol{w}_k - \boldsymbol{w}^* \rangle \\
&= \langle \boldsymbol{s}_{n+1}, \boldsymbol{w}_0 - \boldsymbol{w}^* \rangle - \sum_{k=0}^{n} \eta_k \langle \boldsymbol{g}_k, \boldsymbol{w}_0 - \boldsymbol{w}^* \rangle \\
&= \langle \boldsymbol{s}_{n+1} \circ \boldsymbol{\alpha}_{n+1}^{-1}, (\boldsymbol{w}_0 - \boldsymbol{w}^*) \circ \boldsymbol{\alpha}_{n+1} \rangle \\
&\quad - \sum_{k=0}^{n} \eta_k \langle \boldsymbol{g}_k \circ \boldsymbol{\alpha}_k^{-1}, (\boldsymbol{w}_0 - \boldsymbol{w}^*) \circ \boldsymbol{\alpha}_k \rangle \\
&= \langle \boldsymbol{s}'_{n+1}, (\boldsymbol{w}_0 - \boldsymbol{w}^*) \circ \boldsymbol{\alpha}_{n+1} \rangle - m_{n+1} \\
&= \sqrt{G} \langle \boldsymbol{s}'_{n+1}, \boldsymbol{w}_0 - \boldsymbol{w}^* \rangle - m_{n+1}
\end{aligned} \tag{19}
$$

Applying the definition of $\hat{d}_k$ yields:

$$\hat{d}_{n+2} = \frac{m_{n+1}}{\|s'_{n+1}\|} \le D\sqrt{G} \tag{20}$$

As $d_k$ is non-decreasing and bounded, it converges. Consequently, $\lambda_k = d'_k / \|\boldsymbol{g}'_M\|$ also converges. Because $\lambda_k$ converges to $\lambda$, $\lambda_k > \lambda/2$ for $k > m$.

$$
\begin{aligned}
&\|(\boldsymbol{w}_{k+1} - \boldsymbol{w}^*) \circ \boldsymbol{\alpha}_k\|^2 \\
&= \|(\boldsymbol{w}_k - \boldsymbol{w}^*) \circ \boldsymbol{\alpha}_k - \eta_k \boldsymbol{g}_k \circ \boldsymbol{\alpha}_k^{-1}\|^2 \\
&= \|(\boldsymbol{w}_k - \boldsymbol{w}^*) \circ \boldsymbol{\alpha}_k\|^2 + \eta_k^2 \|\boldsymbol{g}_k \circ \boldsymbol{\alpha}_k^{-1}\|^2 - 2\eta_k \langle \boldsymbol{g}_k, \boldsymbol{w}_k - \boldsymbol{w}^* \rangle \\
&\le \|(\boldsymbol{w}_k - \boldsymbol{w}^*) \circ \boldsymbol{\alpha}_k\|^2 + \eta_k^2 \|\boldsymbol{g}_k \circ \boldsymbol{\alpha}_k^{-1}\|^2 - 2\eta_k (f(\boldsymbol{w}_k) - f(\boldsymbol{w}^*))
\end{aligned} \tag{21}
$$

$$
\begin{aligned}
&\lambda \sum_{k=m}^{n} \gamma_k (f(\boldsymbol{w}_k) - f(\boldsymbol{w}^*)) \\
&\le 2 \sum_{k=m}^{n} \eta_k (f(\boldsymbol{w}_k) - f(\boldsymbol{w}^*)) \\
&\le \|(\boldsymbol{w}_m - \boldsymbol{w}^*) \circ \boldsymbol{\alpha}_m\|^2 - \|(\boldsymbol{w}_{n+1} - \boldsymbol{w}^*) \circ \boldsymbol{\alpha}_{n+1}\|^2 + 2\sum_{k=m}^{n} \eta_k^2 \|\boldsymbol{g}_k \circ \boldsymbol{\alpha}_k^{-1}\|^2 \\
&\quad + \sum_{k=m}^{n} \|(\boldsymbol{w}_{k+1} - \boldsymbol{w}^*) \circ \boldsymbol{\alpha}_{k+1}\|^2 - \|(\boldsymbol{w}_{k+1} - \boldsymbol{w}^*) \circ \boldsymbol{\alpha}_k\|^2
\end{aligned} \tag{22}
$$

The last term is bounded as follows.

$$\sum_{k=m}^{n} \|(\boldsymbol{w}_{k+1} - \boldsymbol{w}^*) \circ \boldsymbol{\alpha}_{k+1}\|^2 - \|(\boldsymbol{w}_{k+1} - \boldsymbol{w}^*) \circ \boldsymbol{\alpha}_k\|^2$$

$$= \sum_{k=m}^{n} \langle (\boldsymbol{w}_{k+1} - \boldsymbol{w}^*) \circ (\boldsymbol{\alpha}_{k+1} + \boldsymbol{\alpha}_k), (\boldsymbol{w}_{k+1} - \boldsymbol{w}^*) \circ (\boldsymbol{\alpha}_{k+1} - \boldsymbol{\alpha}_k) \rangle$$

$$\leq \sum_{k=m}^{n} \|(\boldsymbol{w}_{k+1} - \boldsymbol{w}^*) \circ (\boldsymbol{\alpha}_{k+1} + \boldsymbol{\alpha}_k)\| \|(\boldsymbol{w}_{k+1} - \boldsymbol{w}^*) \circ (\boldsymbol{\alpha}_{k+1} - \boldsymbol{\alpha}_k)\|$$

$$\leq 2D\sqrt{G} \sum_{k=m}^{n} \|(\boldsymbol{w}_{k+1} - \boldsymbol{w}^*) \circ (\boldsymbol{\alpha}_{k+1} - \boldsymbol{\alpha}_k)\| \leq 2D\sqrt{G}D_\infty \sum_{k=m}^{n} \|\boldsymbol{\alpha}_{k+1} - \boldsymbol{\alpha}_k\|_1 \leq B \quad (23)$$

Therefore, since $\|\boldsymbol{g}_k \circ \boldsymbol{\alpha}_k^{-1}\| < \sqrt{G}$,

$$\min_{m+1 \leq k \leq n} f(\boldsymbol{w}_k) - f(\boldsymbol{w}^*) \leq \frac{\sum_{k=m}^{n} \gamma_k (f(\boldsymbol{w}_k) - f(\boldsymbol{w}^*))}{\sum_{k=m}^{n} \gamma_k}$$

$$\leq \frac{\|(\boldsymbol{w}_m - \boldsymbol{w}^*) \circ \boldsymbol{\alpha}_m\|^2}{\lambda \sum_{k=m}^{n} \gamma_k} + \frac{2\lambda^2 G \sum_{k=m}^{n} \gamma_k^2}{\sum_{k=m}^{n} \gamma_k} + \frac{B}{\lambda \sum_{k=m}^{n} \gamma_k} \quad (24)$$

