# OpenReview forum: "Interpreting Adaptive Gradient Methods by Parameter Scaling for Learning-Rate-Free Optimization"
_ICLR.cc/2024/Conference — Submitted to ICLR 2024_

### Official Review · Reviewer_u1Lz · 2023-10-31

**Soundness:** 2 fair
**Presentation:** 2 fair
**Contribution:** 2 fair
**Rating:** 5
**Confidence:** 2

**Summary:**

This paper proposes a method to apply adaptive gradient methods, such as Adam, to learning-rate-free methods for deep learning.
The experiments demonstrate the proposed method works on various scenarios, including image classification to reinforcement learning and semantic segmentation.

**Strengths:**

The experiments are conducted on cases where learning rate configuration is crucial, such as reinforcement learning and training of ViT from scratch and demonstrate that the approach is comparable or even better to the baselines.
I hope this approach relieve us from learning rate tuning.

**Weaknesses:**

* To my understand $c$ in Algorithm 1 is a hyperparameter. If so, does it mean that this method introduces a parameter to eliminate learning rate? How sensitive the proposed method to this parameter?
* Algorithm 1 also requires $f^*$, which I think is the loss value at the optimum. For deep models, obtaining such a value sounds quite challenging.

**Questions:**

* How to tune $\gamma_k$?

---

> ### Author Response · Authors · 2023-11-11
>
> W1) We apologize for the confusion. Below is our response to weakness 1:
>
> Although Alg. 1 introduces a hyperparameter, $c$, it demonstrates robustness to this hyperparameter. Specifically, we consistently employed $c=0.5$ across all experiments, including supervised classification, self-supervised learning, fine-tuning, and reinforcement learning.
>
> &nbsp;
>
> W2) Yes. $f^*$ represents the loss value at the optimum, and achieving this value can be challenging in deep learning applications. However, we can simply use the theoretically minimum value of the loss function as $f^*$. For instance, we set $f^*=0$ for cross-entropy losses and the SimCLR loss. On the other hand, since we cannot determine the minimum loss values for reinforcement learning models, we have omitted the experiments in Table 3. (SPS and PS-SPS in reinforcement learning tasks)
>
> &nbsp;
>
> Q1) $\gamma_k$ is an optional hyperparameter for learning rate annealing. We can simply employ the same learning rate annealing schedule used in previous methods, such as a step decay or a cosine annealing. It is worth noting that the annealing should have a base learning rate of $1$. For example, a step decay should start from $1$ and decay to $0.1$ and $0.01$.

---

> > ### Comment · Reviewer_u1Lz · 2023-11-22
> >
> > Thank you for the comments.
> >
> > * W1
> > > we consistently employed  $c=0.5$
> >
> > Along with the reply to Q1 that you use annealing, I got uncertain how to see your empirical results.
> > I think one can train neural networks very well by using Adam, for example, with a default learning rate with a (careful) annealing strategy.
> > The difference between using learning-rate-free methods and Adam, or things like this, is unclear to me.
> >
> > * W2
> > Thank you for the clarification. It would be appreciated if you state it clearly in the manuscript.

---

> > > ### Author Response · Authors · 2023-11-22
> > >
> > > We would like to emphasize that SPS amd PS-SPS (Alg. 1) do not employ an additional annealing strategy ($\gamma_k$), apart from D-Adaptation-based methods like D-Adapt SGD, D-Adapt Adam, Prodigy, and PS-DA-SGD (Alg. 2), which include $\gamma_k$.
> > >
> > > To ensue a fair comparison, we applied the same annealing schedule used for training baseline models. Specifically, we used a step decay in CIFAR-100 experiments, a constant learning rate in reinforcement learning models, and a cosine annealing in ImageNet, fine-tuning, and SSL. We will provide further clarification on this point in the Appendix.
> > >
> > > It is true that a careful annealing strategy enables an optimizer to converge with arbitrary learning rates. For instance, in Fig. 1 (a), Prodigy exhibits a sudden increase in the learning rate, which can be handled by a rapidly decreasing schedule. However, we did not further tuned the schedule and utilized the ones provided by baseline codes.

---

### Official Review · Reviewer_jkmf · 2023-10-31

**Soundness:** 3 good
**Presentation:** 3 good
**Contribution:** 3 good
**Rating:** 8
**Confidence:** 4

**Summary:**

The paper proposes an efficient learning-rate free gradient-descent type optimization technique. The approach reconciles learning-rate-free approaches with parameter-wise adaptive gradient scaling methods. This result is achieved intuitively by reinterpreting gradient scaling as parameter rescaling. The approach builds on recently introduced methods for learning-rate-free optimization techniques and extends those to a parameter-wise step-size adaptation.

**Strengths:**

The presentation of the paper is clear and the approach is simple yet original and efficient and has potentially a promising impact.

**Weaknesses:**

While the approach is intuitive and a convergence proof is given, the approach exhibits heuristic qualities and doesn't discuss the resulting dynamic of the adaptation. Especially, in consideration of the potential complex resulting dynamics by applying parameter wise step-size adaptation.

**Questions:**

Is there something that can be said about label noise sensitivity/robustness of the proposed method?

---

> ### Author Response · Authors · 2023-11-12
>
> W1) We apologize for the confusion. Our proposed method estimates one step-size for one network.
>
> For example, when employing Adam to train a network, there exists one step-size hyperparameter, and our method is designed to estimate this value.
>
> The parameter-wise scaling of the proposed method corresponds to the gradient scaling of Adam, where the gradient of each parameter is divided by the square root of the moving average of the second moment of the gradient. Similarly, the proposed method scales each parameter by the fourth root of the moving average of the second moment of the gradient. The effects of these scaling are equivalent as shown in Eqs. 1-4.
>
> &nbsp;
>
> Q1) Thank you for the insightful suggestion. To measure the label noise sensitivity of the proposed method, we conducted an additional experiment using the ResNet-32 network on the CIFAR-100 dataset. To introduce label noise, we randomly shuffled some labels in the training dataset. The Adam optimizer with a learning rate of $0.003$ was chosen as the baseline, which is the optimal learning rate value found by grid-search on the dataset without label noise.
>
> | Label noise | 0.0   | 0.2   | 0.4   | 0.6   | 0.8   |
> |-------------|-------|-------|-------|-------|-------|
> | Adam        | 66.31 | 50.95 | 39.75 | 25.10 | 9.65  |
> | PS-DA-SGD   | 64.49 | 52.57 | 41.34 | 25.26 | 10.41 |
>
> The proposed method demonstrates robustness to the label noise by surpassing the baseline when the label noise is injected. It is due to the optimal learning rate depends on the level of label noise. We used a learning rate of $0.003$ for Adam across all label noise levels. While this value was the optimal value in the absence of label noise, it was suboptimal when the level of label noise changed. Contrarily, the proposed method found the optimal learning rate regardless of the label noise level.

---

### Official Review · Reviewer_AxR5 · 2023-10-31

**Soundness:** 2 fair
**Presentation:** 1 poor
**Contribution:** 1 poor
**Rating:** 3
**Confidence:** 2

**Summary:**

The paper proposes two new algorithms Parameter-scaled stochastic Polyak step-size and Parameter-scaled D-Adapt from the intuition of parameter scaling, and compares their performance to other algorithms.

**Strengths:**

The problem studied is interesting. It is important to learn whether we can make adaptive gradient methods learning rate-free.

**Weaknesses:**

The contribution of the paper is unclear. In the first sentence in the abstract, the authors claim that they "address the challenge of estimating the learning rate for adaptive gradient methods." The issue is important, but after reading the paper, I do not follow how they addressed the issue.

Then, the authors claim they "interpret adaptive gradient methods as steepest descent applied on parameter-scaled networks ." Authors need to explain why their new interpretation is important to the ICLR community.

Also, the authors claim they "propose learning rate-free adaptive gradient methods". It appears that algorithm 2 is the method they propose. However, in algorithm 2, there are a lot of hyper-parameters, including $\eta_k$, $\gamma_k$, and even $\alpha_k$. It is not clear to me why Algorithm 2 is "learning rate-free". The explanation about the notations in Algorithm 2 should be clearer.

In section 5.2, authors should report the metrics for their reinforcement learning experiment. It is hard to understand the value of the proposed PS-SPS and PS-DA-SGD from Table 3. Also, the authors mention that they removed all the batch normalization layers in the CIFAR-100 experiment. What are the benefits of such removal?

Authors should also clearly write their assumptions and conclusions into a formal theorem in Section 4.2.

**Questions:**

Please see the weakness section.

---

> ### Author Response · Authors · 2023-11-14
>
> We apologize for any confusion. We have clarified the statements in our paper as follows:
>
> W1) In our paper, we present two algorithms to estimate the learning rate of adaptive gradient methods. In Alg. 1, line 6 represents the update rule of an optimizer, with the learning rate $\eta_k$ estimated in line 5. Similarly, in Alg. 2, line 11 represents the update rule of an optimizer, with the learning rate $\eta_k$ estimated in line 10.
>
> &nbsp;
>
> W2) Our proposed interpretation of adaptive gradient methods enables the transformation of learning-rate-free optimizers developed for steepest descent [1], [2] into adaptive gradient learning-rate-free optimizers.
>
> &nbsp;
>
> W3) Training a deep neural network with an adaptive gradient optimizer involves various design choices:
> 1. Selecting a gradient rescaling strategy, such as AdaGrad, Adam, or AMSGrad.
> 2. Choosing a base learning rate within the range of $0.0003$ to $0.01$ or broader.
> 3. Selecting a learning rate annealing schedule, such as constant, step decay, or cosine annealing.
>
> In this context, Alg. 1 relieves the second and third design choices, while Alg. 2 addresses the second design choice. These algorithms are termed learning-rate-free methods as they alleviate the manual tuning of the base learning rate.
>
> The $\alpha_k$ term in Alg. 1 and Alg. 2 corresponds to the first design choice. For example, if we choose Adam as the optimizer, then $\alpha_k$ will be $v_k^{1/4}$ (see Table 1 and Eq. 4). The $\gamma_k$ of Alg. 2 corresponds to the third design choice, representing the learning rate annealing schedule. For example, if we choose a step decay schedule, then $\gamma_k$ will start from $1$ and decay by a factor of $10$.
>
> The remaining parameters, $c$ of Alg. 1 and $d_0$ of Alg. 2, are hyperparameters. However, the algorithms are robust to their values. We used $c=0.5$ and $d_0=10^{-6}$ across all experiments, including supervised classification, self-supervised learning, fine-tuning, and reinforcement learning.
>
> &nbsp;
>
> W4) In our experiments evaluating reinforcement learning, we utilized the average of the reward output of the OpenAI Gym environment as the performance metric. The values in parentheses represent the training success rate. It is important to note that SPS and PS-SPS were omitted from these experiments due to their reliance on the loss value at optimum, denoted as $f^*$, which is challenging to estimate before training within the context of reinforcement learning models.
>
> We have removed all batch normalization layers from VGG networks, while ResNet networks remain unchanged. This modification aims to assess the impact of batch normalization on learning-rate-free algorithms. Our rationale for investigating this impact of batch normalization stems from the observation that many learning-rate-free methods exhibit subpar performance in reinforcement learning experiments, where batch normalization is not employed.
>
> &nbsp;
>
> W5) A formal theorem of Sec. 4.2 is as follows.
>
> AMGRrad scaling rule:
>
> $\alpha_k=(\max_{i\le k}v_i)^{1/4}$, where $v_{i+1}=\beta v_i + (1 - \beta) g^2_i$, $0<\beta<1$.
>
> For a convex G-Lipschitz function $f$ and AMSGrad scaling rule, Alg. 1 converges to the minimum of $f$.
>
> For a convex G-Lipschitz function $f$, AMSGrad scaling rule, and a decreasing sequence $\gamma_k$ satisfying $\sum^\infty_{k=1}\gamma_k=\infty$ and $\sum^\infty_{k=1}\gamma_k^2<\infty$, Alg. 2 converges to the minimum of $f$.
>
> &nbsp;
>
> [1] Stochastic Polyak Step-size for SGD: An Adaptive Learning Rate for Fast Convergence, Loizou, Nicolas, et al., AISTATS (2021)
>
> [2] Learning-Rate-Free Learning by D-Adaptation, Aaron Defazio, Konstantin Mishchenko, ICML (2023)

---

### Official Review · Reviewer_jSjZ · 2023-11-04

**Soundness:** 2 fair
**Presentation:** 2 fair
**Contribution:** 2 fair
**Rating:** 5
**Confidence:** 3

**Summary:**

This paper proposes a learning rate tuning method for adaptive optimization algorithms. Besides, this paper also proposes a method to interpret adaptive gradient methods as parameter-scaled SGD. Experimental results show that the proposed method can be comparable with adaptive gradient methods with hand-tunned learning rates.

**Strengths:**

1. This paper uses parameter rescaling to interpret adaptive gradient methods, which could be helpful for further investigating the behavior of adaptive gradient methods.
2. The proposed learning-rate-free methods can be useful to avoid the hyperparameter tunning in adaptive gradient methods while still achieving fast convergence.

**Weaknesses:**

1. The paper organization is not clear to me. In particular, Algorithm 2 looks pretty complicated to me. The authors just explain each steps after the algorithm, while I am still not very clear about the motivation and why such a method can be developed.

2. Second, the equations are also not clear. The authors claim that the adaptive methods can be viewed as applying the steepest descent to parameter-scaled networks based on Eqs 1-4. However, the notations are not clear, what's the formal definition of $f'$, why $f'$ needs to be introduced, and how to leverage it?

3. In Section 3, equation (7) is also confusing, in Adam, $\alpha$ is also depending on the randomness of the stochastic gradients, when why $E u = \nabla f(w)/\alpha^2$ can hold?

4. The reasoning from (9)-(11) is also not clear to me, if you only want to mention that the learning rate should not depend on $\alpha$, why do you still need equations (10) and (11)?

5. The convergence analysis is also not clear, the authors just provide a very simple proof in the appendix, in the main part, I actually do not see anything that is related to the convergence. Additionally, the proof is also not clear, many notations such as D,G are not clearly presented; the assumption that $\alpha_k$ coverges to $\alpha$ is also not presented; Eq. (15) is also not well justified.

**Questions:**

Please see the weakness section.

---

> ### Author Response · Authors · 2023-11-11
>
> W2) Sorry for the ambiguous explanation. Below is our response to weakness 2:
>
> Given a locally convex function $y=f(w)$, an optimizer tries to find its local minimum $w_*$ with an initial guess of $w_0$. Here, we stretch $f$ in the $w$-axis by a factor of $\alpha$. Therefore, the formal definition of $f'$ is given as $f'(w)=f(w/\alpha)$. [Eq. (1)]
>
> The reason we introduce $w'$ is to emphasize the relationship between the local minimum and the initial guess of $f'$ and $f$, where the local minimum of $f'$ becomes $\alpha w_* = w'_*$ and our initial guess becomes $\alpha w_0 = w'_0$.
>
> The above observation leads to our main idea: stretching $f$ by $\alpha$ **increases** the distance to the solution by the factor of $\alpha$, but the gradient **decreases** by the factor of $\alpha$. Therefore, if we use a learning rate of $\eta$ to find the local minimum on $f'$, its behavior will be equivalent to using a learning rate of $\eta/\alpha^2$ to find the local minimum on $f$. [Eqs. (2-4)]
>
> In the above discussion, we focused on the case of a scalar-valued $\alpha$. However, the same principles can be applied to transform adaptive gradient methods into steepest descent by stretching $f$ with different values for each axis.
>
> &nbsp;
>
> W4) Sorry for the confusion. Below is our response to weakness 4:
>
> While the learning rate may depend on $\alpha$, we would like to emphasize that the factor that should not depend on $\alpha$ is the trajectory of parameter. Assume an optimizer updates the parameter of $f$ with $w_0$, $w_1$, $w_2$, ..., $w_t$ and updates $f'$ with $w'_0$, $w'_1$, $w'_2$, ..., $w'_t$. Here, we used the stretched function $f'(w)=f(w/\alpha)$. If a learning-rate-free optimizer actually estimates the optimal learning rate, then the trajectory should be identical regardless of $\alpha$, i.e., $w'_i=w_i/\alpha$ for all $i$.
>
> However, without employing our parameter scaling approach, simply scaling gradients while leaving parameters intact (which is the case of previous adaptive gradient methods) results in trajectories that depends on $\alpha$. [Eqs. (9-11)]
>
> In contrast, it can be easily shown that our parameter scaling approach ensures identical trajectories as following. In this case, we estimate the learning rate on $\alpha$-stretched $f’(w’)$, which means $f'(w')=f(w)$, $\nabla f'(w')=\nabla f(w)/\alpha$, and $w’=\alpha w$. Therefore, $E[\eta u]$, which means the estimated step, becomes proportional to $\alpha$. Since $\alpha$-stretched function leads to an estimated step that is proportional to $\alpha$, out parameter scaling approach ensures identical parameter trajectories across different scalings.
>
> &nbsp;
>
> W3) Thank you for pointing out the ambiguity. Below is our response to weakness 3:
>
> Eq. 7 holds true only when the coefficient for computing the moving average of second moment (denoted as $\beta_2$ in the case of Adam) is sufficiently large so that the stochasticity of gradient can be ignored.

---

> ### Author Response · Authors · 2023-11-11
>
> W1) Sorry for the insufficient explanation. We appreciate the opportunity to further provide details about Alg 2.
>
> Alg. 2 describes the procedures for transforming D-Adapt SGD [1] into a form suitable for adaptive gradient methods. The main challenge we encountered during this transformation arose from its method of estimating learning rate, denoted as $\eta=\frac{d \gamma}{\lVert g_0 \rVert}$ [Alg. 4 of [1]].
>
> In adaptive gradient methods, $\alpha$ is decreased to ensure the consistent gradient scaling throughout training. However, when a parameter converges earlier than others, the corresponding element of $\alpha$ approaches zero, which hinders the straightforward application of our parameter scaling approach. If we scale parameters using $\alpha$, one of whose elements is near zero, $\lVert g_0' \rVert=\lVert g_0/\alpha\rVert$ diverges. Consequently, $\eta$ becomes zero, leading to early termination of the training process if we directly apply the parameter scaling approach to D-Adapt SGD.
>
> To address the risk of early termination, we opted to scale gradients using $\alpha_M$ instead of $\alpha_k$. [line 6, 9]
>
> However, this modification introduces an issue -- specifically, it no longer ensures the identical trajectories of parameter across different scalings, which is described in our response to weakness 4. To mitigate this concern, we introduced line 7. In the D-Adaptation algorithm, $d$ represents the estimated lower bound of the distance to solution from the initial guess. Given that the distance to solution is also influenced by the change in parameter scaling, the adjustment in line 7 is justifiable.
>
> Additionally, line 8 provides a minor improvement. In the D-Adaptation algorithm, the Lipshitz constant of $f$, denoted as G, is used to estimate the learning rate. While [1] uses the gradient norm at initial guess as a proxy of G, we utilize the norm of the maximum gradient of each element.
>
> The other lines are either identical to or straightforwardly derived from Alg. 4 of [1].
>
> In contrast, SPS [2] does not suffer from the early termination issue, therefore, does not require extra modifications as outlined in Alg. 1.
>
> &nbsp;
>
> W5) Sorry for the insufficient explanation. Below is our response to weakness 5:
>
> We omit the details about the proof in the main text, because they are simple and constitute only a minor part of our contribution. $G$ denotes the Lipschitz constant of function $f$ and $D$ denotes the distance to optimum from the initial guess. While Alg. 1 and Alg. 2 accept arbitrary scaling factors $\alpha_k$, we employed the AMSGrad [3] scaling rule for the convergence proof, denoted as $\alpha^2_k=\sqrt{\max_{i\le k} v_i}$. As $\alpha_k$ is a non-decreasing sequence with an upper bound of $\sqrt{G}$, it converges to a constant and we denote the constant as $\alpha$.
>
> Further details regarding Eq. 15 are provided below.
>
> If we consider two vectors $x=[x_0, x_1, ..., x_n]$ and  $y=[y_0, y_1, ..., y_n]$ with $x_i \ge y_i$ for all $i$, we can define the element-wise division $x \circ y^-1=[x_0/y_0,...,x_n/y_n]$. Since $x_i/y_i \ge 1$, it follows that $\max (x \circ y^-1) \ge \prod_i x_i / \prod_i y_i$. We define $\beta_k$ as the product of all elements of $\alpha_k$. Since $\alpha_k$ is a bounded vector with a finite number of elements, $\beta_k$ is also bounded. Finally, we define $\beta_n \cdot \beta_m ^{-1}=A^2$.
>
> Additionally, as $\alpha_k$ is a non-decreasing sequence, it follows $\max (\alpha_k \circ \alpha^{-1}_{k-1}) \ge 1$.
>
> &nbsp;
>
> [1] Learning-Rate-Free Learning by D-Adaptation, Aaron Defazio, Konstantin Mishchenko, ICML (2023)
>
> [2] Stochastic Polyak Step-size for SGD: An Adaptive Learning Rate for Fast Convergence, Loizou, Nicolas, et al., AISTATS (2021)
>
> [3] On the Convergence of Adam and Beyond, Reddi, Sashank J., Satyen Kale, and Sanjiv Kumar., ICLR (2018)

---

### Meta-Review · Area_Chair_oNWn · 2023-12-04

**Metareview:**

This paper proposes to interpret the adaptive gradient methods as a parameter-scaled SGD. Based on the proposed interpretation, the authors derived a learning rate tuning method and have carried out several experimental results to show that the proposed method performs comparable hand-tunned method. Though there are big claims like ``we address the challenge of estimating the learning rate for adaptive gradient methods'', the result does not really justify the claim. Moreover, multiple reviewers have complained about the unclear organization of the paper as well as the unclear presentation of the contribution. The AC also read the paper and also find the claims and presentation confusing and unclear. In particular, it has not been explained why interpreting the method as parameter-scaling would help. The lack of explaining insights makes the current interpretation looks more like a technical artifact. For example, explaining GD: $x^k+1 = x^k - \eta\nabla f(x^k)$ as a scaled GD of $f'(x)=f(cx)$ with $\eta/c^2$ step size would not help interpreting the insight of gradient descent. Therefore, the AC suggests a rejection.

**Justification For Why Not Higher Score:**

The theoretical soundness of the paper is limited. The "interpretation" is not fundamental and the AC together with several reviewers are not convince by the claim. Instead, it is more like a technical artifact.

**Justification For Why Not Lower Score:**

N/A

---

### Decision · Program_Chairs · 2024-01-16

Reject